# Discovery and Prevalence of Divergent RNA Viruses in European Field Voles and Rabbits

**DOI:** 10.3390/v12010047

**Published:** 2019-12-31

**Authors:** Theocharis Tsoleridis, Joseph G. Chappell, Elodie Monchatre-Leroy, Gérald Umhang, Mang Shi, Malcolm Bennett, Rachael E. Tarlinton, C. Patrick McClure, Edward C. Holmes, Jonathan K. Ball

**Affiliations:** 1School of Life Sciences, The University of Nottingham, Nottingham NG7 2UH, UK; joseph.chappell@nottingham.ac.uk (J.G.C.); patrick.mcclure@nottingham.ac.uk (C.P.M.); 2Wolfson Centre for Global Virus Infections, The University of Nottingham, Nottingham NG7 2UH, UK; 3Anses, Laboratoire de la Rage et de la Faune Sauvage, 54220 Malzeville, France; elodie.monchatre-leroy@anses.fr (E.M.-L.); gerald.umhang@anses.fr (G.U.); 4Marie Bashir Institute for Infectious Diseases and Biosecurity, School of Life and Environmental Sciences and School of Medical Sciences, The University of Sydney, Sydney, NSW 2006, Australia; mang.shi@sydney.edu.au (M.S.); edward.holmes@sydney.edu.au (E.C.H.); 5School of Veterinary Medicine and Science, University of Nottingham, Sutton Bonington Campus, Loughborough LE12 5RD, UK; m.bennett@nottingham.ac.uk (M.B.); rachael.tarlinton@nottingham.ac.uk (R.E.T.)

**Keywords:** virus discovery, paramyxovirus, rotavirus A, astrovirus, picorna-like virus, rodents, rabbits, field voles

## Abstract

The advent of unbiased metagenomic virus discovery has revolutionized studies of virus biodiversity and evolution. Despite this, our knowledge of the virosphere, including in mammalian species, remains limited. We used unbiased metagenomic sequencing to identify RNA viruses in European field voles and rabbits. Accordingly, we identified a number of novel RNA viruses including astrovirus, rotavirus A, picorna-like virus and a narmovirus (paramyxovirus). In addition, we identified a sobemovirus and a novel luteovirus that likely originated from the rabbit diet. These newly discovered viruses were often divergent from those previously described. The novel astrovirus was most closely related to a virus sampled from the rodent-eating European roller bird (*Coracias garrulous*). PCR screening revealed that the novel narmovirus in the UK field vole had a prevalence of approximately 4%, and shared common ancestry with other rodent narmoviruses sampled globally. Two novel rotavirus A sequences were detected in a UK field vole and a French rabbit, the latter with a prevalence of 5%. Finally, a highly divergent picorna-like virus found in the gut of the French rabbit virus was only ~35% similar to an arilivirus at the amino acid level, suggesting the presence of a novel viral genus within the *Picornaviridae*.

## 1. Introduction

RNA viruses likely infect every species of cellular life [1]. However, due to their potential and sometimes profound impact on public health and the agricultural industries, most attention has understandably been directed toward those viruses that affect humans and economically important animals and plants. Hence, our knowledge of the diversity, evolution and functional biology of RNA viruses infecting a wider range of host species is limited. This major sampling bias has been in part addressed by recent studies reporting the discovery of a multitude of new invertebrates and vertebrate viruses, massively increasing the known virosphere and demonstrating the importance of investigating species that are often overlooked [2,3,4,5,6,7,8].

Rodents are an important source of emerging virus infections [9] and are reservoir hosts for a wide range of viruses including coronaviruses, paramyxoviruses and picornaviruses [10,11,12,13]. The abundant and diverse virome harbored by rodents likely reflects their propensity to live in relatively large and dense populations and their speciose nature: Rodentia is the largest mammalian order, comprising 2200 species, and hence approximately 40% of all mammalian species [14,15]. Lagomorphs are closely related to rodents, although comprise only 89 extant species including rabbits and hares [16]. Recent reports have shown that rabbits and hares harbor an array of viruses such as caliciviruses, coronaviruses, bocaparvoviruses, hepatitis E and myxomavirus [17,18,19,20,21,22]. Importantly, the discovery of these novel viruses in rodents and lagomorphs has provided a more refined understanding of the evolutionary history of known pathogenic viruses, including coronaviruses, hepatitis A virus and hepaciviruses [23,24,25,26,27]. However, despite these recent advances, we have only scratched the surface of the potential diversity of rodent and lagomorph viruses.

In this study, we generated and investigated RNA sequencing data from European rodents and rabbits that were previously found to be positive for novel alphacoronaviruses and hantaviruses [11,20,25,28]. Our aim was to perform an unbiased discovery of additional viruses that were present in these samples, reveal their evolutionary history, and assess their prevalence in their host populations. Accordingly, we demonstrate the presence of novel and highly divergent RNA viruses and discuss the potential pitfalls of the use of enteric samples for virus discovery.

## 2. Materials and Methods 

### 2.1. Samples

We re-analyzed two samples previously reported to harbor alphacoronaviruses [11,20,25] and one sample positive for hantavirus [28]. These samples included a UK *Microtus agrestis* gut (UK*Ma1*), a UK *Microtus agrestis* kidney (UK*Ma K4D*) and a France-resident *Oryctolagus cuniculus* intestinal wash (L232) (Table 1).

Additional screening for paramyxoviruses was performed on 130 field vole kidney samples collected from two locations in the UK (Cheshire and Leicestershire (Figure 1A)) [11,28], using primers specific for the novel paramyxovirus identified in UK*Ma* K4D. The primers were targeting the L gene F: 5′-ACTAATTACATTTACCAACAAGG-3′ and R: 5′-AGTTGGTCACGTTCWATAAT-3′ and generated a 245 bp PCR product.

Moreover, screening with primers specific for the rotavirus A that was identified in the French rabbit L232 was performed on 268 rabbit intestinal wash samples collected from 10 departments of France (North-West: Finistère, Morbihan, Loire Atlantique. Centre: Creuse, Charente, Puy-de-Dôme, Deux-Sèvres, Loire and Dordogne. South-West: Pyrénées Orientales (Figure 1B)) [20]. The primers targeted the *NSP4* segment F: 5′-TGAAGATCCAGGAATGGCGT-3′ and R: 5′-ACCTGCCAACTTTAATAGCGT-3′ generating a 178 bp PCR product.

All the samples used in this study were sourced from existing pest controls programs and approved by the University of Nottingham School of Veterinary Science Ethical Panel, reference numbers 1602 151102 and 1786 160518.

### 2.2. Nucleic Acid Preparation

Total RNA was extracted from 1 mm^3^ sections of kidney (UK*Ma* K4D) or intestinal tissue samples (UK*Ma*1) using the GenElute^™^ Mammalian Total RNA Miniprep Kit (Sigma Aldrich, Steinheim, Germany). Total RNA was extracted from intestinal fluid for the French L232 rabbit using the QIAamp Viral RNA Mini Kit (Qiagen, Hilden, Germany). The RNA to cDNA EcoDry™ Premix—Random Hexamers kit (Clontech, Saint-Germain-en-Laye, France)—was used for cDNA synthesis.

### 2.3. High-Throughput Sequencing

High-throughput sequencing was performed on total RNA extracted from all the three samples (UK*Ma* K4D, UK*Ma*1 and L232) using the Illumina HiSeq platform at Source Bioscience, Nottingham, UK. Genomic DNA (gDNA) was depleted using DNaseI, and ribosomal RNA (rRNA) was removed using NEBNext^®^ rRNA Depletion Kit (Human/Mouse/Rat) with RNA Sample Purification Beads (New England Biolabs, Ipswich, MA, USA) prior to the library construction. Each read length was 2 × 150 bp, and the insert size was 200 bp on average. All the sequence data generated were analyzed using the Geneious Prime 2019.0.4 software. The reads were assembled into contiguous sequence (contigs) with de novo assembly performed with Geneious Prime. By using BlastX, the assembled contigs were compared against the RefSeq sequence database of all the virus proteins downloaded from GenBank. The minimum e-value was set to 1 × 10^−5^ to maintain both high sensitivity and low rates of false positive hits [2]. All the contigs that matched viruses were confirmed by PCR.

### 2.4. Analysis of Virus Abundance

The number of viral reads in each data set was measured by mapping the raw total reads generated from the high-throughput sequencing to the identified viral contigs. The host *ribosomal protein L4 (RPL4)* gene was used as a host gene reference marker. The *Oryctolagus cuniculus RPL4* gene (NM_001195817.1) was used for the rabbit, while the *Microtus ochrogaster RPL4* gene (XM_005347781.3) was employed for the field vole. The abundance of viral reads was calculated by dividing the number of mapped reads by the total number of reads in each library. The graphs were generated with GraphPad Prism 8 (v8.1.2, GraphPad Software, San Diego, California USA).

### 2.5. Contig Confirmation PCR

Specific primers were designed for contig confirmation, and for gap-filling between contigs, based on the known sequences of the contigs. All the primers were evaluated in silico with the Primer3 online tool. The Geneious Prime software was used for mapping and annotating. PCR reactions were carried out in a PTC-200 Peltier Thermal Cycler (MJ Research, Waltham, Massachusetts, USA). The PCR reactions were performed with HotStarTaq Polymerase (Qiagen, Hilden, Germany) according to manufacturer’s instructions. The PCR products were subjected to Sanger sequencing at Source Bioscience, Nottingham.

All the virus sequences generated in this study have been deposited on GenBank under the accession numbers MN626413-MN626440.

### 2.6. Phylogenetic Analysis

To facilitate phylogenetic analysis, 57 reference paramyxovirus sequences were downloaded from GenBank and the *F*, *H*, *L*, *M*, *N* and *P* genes extracted. Similarly, 50 rotavirus A reference sequences were downloaded for the VP1, VP2, VP3, VP6, NSP2, NSP3 segments, as were 47 reference sequences of astroviruses for *ORF1a, ORF1b* and *capsid* genes. Amino acid and nucleotide sequences were aligned using ClustalW [29] and phylogenetic analyses were performed using the maximum likelihood (ML) method within the Molecular Evolutionary Genetics Analysis version 7 (MEGA7) [30] package. Analyses of aligned amino acid sequences utilized a Jones–Taylor–Thornton (JTT) amino acid substitution model with uniform rates of variation, and complete deletion of gaps, with statistical robustness assessed using bootstrap resampling (500 pseudo-replicates). In addition, a phylogenetic analysis of nucleotide sequence data was performed using the General Time Reversible (GTR) nucleotide substitution model with a gamma distribution of rate variation, a class of invariant sites (Γ+I), and complete deletion of gaps, with statistical robustness assessed using bootstrap resampling (100 pseudo-replicates). As the nucleotide-based phylogenies (Appendix A) were topologically similar to those inferred using amino acid sequences only the latter are shown in the main text.

Details on the number of sequences used for the phylogenetic analysis of each gene as well as their length are given in Table 1.

## 3. Results

### 3.1. Abundance of Viral Reads

A total of 51,744,718 paired reads were generated for UK*Ma* K4D, 63,152,134 for UK*Ma*1, and 8,396,334 for L232. The rabbit L232 (intestinal wash) had the highest abundance of viral reads ranging between 0.0008% and 0.19%, followed by the field vole UK*Ma* K4D (kidney) with 0.0003–0.002% and finally the field vole UK*Ma*1 (intestine) with 0.00004–0.0002% (Figure 2A,B). The number of host *RPL4* reads was proportional to the number of viral reads in each sample. The abundance of viral and *RPL4* reads in each animal were correlated: the animals with the highest abundance of host reads also had the highest abundance of viral reads. A novel rotavirus A, picorna-like virus and coronavirus (previous study [20]) were identified in the rabbit, a novel paramyxovirus and hantavirus (previous study [28]) in the field vole UK*Ma* K4D, and a novel astrovirus, rotavirus A and coronavirus (previous study [11,25]) in the field vole UK*Ma*1 (Figure 2A,B). We now briefly describe each in turn.

### 3.2. Paramyxovirus

A novel narmovirus was discovered in the kidney of the UK field vole (UK*Ma* K4D). A total of 6734 nt were retrieved for this virus, representing the *F*, *H*, *L*, *M*, *N* and *P* genes (Table 1). In terms of sequence similarity, its closest relative was Bank vole virus (MF943130), with 76%, 61%, 71%, 87%, 90% and 62% similarity for *F*, *H*, *L*, *M*, *N* and *P* genes, respectively, at the amino acid level (Table 2). 

To determine the evolutionary relationship of the novel UK field vole paramyxovirus to the other viruses within the family, we performed phylogenetic analyses for each of the *F*, *H*, *L*, *M*, *N* and *P* genes. The same phylogenetic pattern was observed in every gene, with the rodent narmoviruses comprising a distinct clade within the family with >90% bootstrap support (Figure 3, Appendix A). Specifically, this rodent-specific clade comprised UK*Ma* K4D, Bank vole virus (MF943130), Mossman virus (NC005339) and Nariva virus (NC017937), and has been named *Narmovirus* genus within the Paramyxoviridae by ICTV.

Further PCR screening on additional UK field vole samples showed 3.8% prevalence (5 positives in 130 voles tested) (Figure 1A).

### 3.3. Rotavirus A

Two rotaviruses A were discovered in the intestinal samples of the UK field vole (UK*Ma*1) and the French rabbit L232. A total of 7565 nt and 13,278 nt (Table 1) were retrieved for UK*Ma*1 and L232, respectively. Blastn analysis of all the available rotavirus A segments (VP1, VP2, VP3, VP6, VP7, NSP2, NSP3, and NSP4) revealed that the rabbit RVA was closely related (>96%) to other viruses from a range of host species including human, bovine, giraffe and roe deer (Table 2). These findings were confirmed by the phylogenetics analysis for each segment (Figure 4, Appendix A). In contrast, the field vole rotavirus A was more divergent, with nucleotide similarities of ~75% to the closest relatives in the VP1, VP2, VP3, VP6, NSP2 and NSP3 segments, suggesting a novel genotype of RVA in each of the retrieved segments (Table 2). Phylogenetic analysis showed that UK*Ma*1 RVA formed a distinct lineage within rotavirus A species, although it loosely grouped with a mouse RVA in the VP3 phylogeny (Figure 4 and Appendix A). Detailed conclusions regarding the evolutionary history of each segment of the rabbit and field vole RVA could not be drawn due to low bootstrap support on the branches of each tree. Additional PCR screening on more French rabbit intestinal washes from the same cohort revealed a 4.9% prevalence (13 positives in 268 samples). Six originated from Creuse department in central France, four from the neighboring Puy-de-Dôme department, with the other three from the departments Deux-Sèvres, Loire and Dordogne (Figure 1B). All the positive samples originated from central France with the exception of Charente, whereas no positive cases were identified in the north-west and south-west departments. However, no more positives were detected in the UK rodent cohort.

### 3.4. Astrovirus

A novel mamastrovirus was discovered in the intestine of the UK field vole (UK*Ma*1). In total, 4334 nt were retrieved for the *ORF1a*, *ORF1b* and the *capsid* genes (Table 1). BlastX results revealed that the closest relative was Astrovirus Er/SZAL6/HUN/2011 from a European roller bird (*Coracias garrulus*). The two viruses were 46%, 70% and 73% similar at the amino acid level in the *ORF1a*, *ORF1b* and *capsid* genes, respectively (Table 2). Phylogenetic analysis of all the three genes showed that the field vole and roller viruses formed a distinct clade within the Mamastrovirus genus supported by >90% bootstrap replicates in capsid and ORF1a (Figure 5 and Appendix A).

### 3.5. Picorna-Like Virus

Two contigs of a novel picorna-like virus were retrieved from the intestine of the French rabbit L232 with a total of 5642 nt (4752 nt and 890 nt) (Table 1). A blastX analysis revealed that in the NTPase, 3C peptidase, RNA-dependent-RNA-polymerase (RdRp) and capsid proteins the novel rabbit picorna-like virus was only 44%, 52%, 61% and 34% similar at the amino acid level to their closest relative (Blackbird arilivirus) (NC040820) (Table 2). Phylogenetic analysis revealed that the virus clustered with the Blackbird arilivirus with 100% bootstrap support (Figure 6). Surprisingly, both viruses clustered within the invertebrate picorna-like viruses, suggesting that they might have in fact been derived from a dietary component.

### 3.6. Plant Viruses

In addition to the vertebrate and invertebrate viruses described above, two likely plant viruses were detected in the intestinal wash of the French rabbit: a sobemovirus and a luteovirus. A 1653 nt contig was retrieved for the sobemovirus, encompassing polyprotein P2ab (Table 1). The polyprotein P2ab was both 98% similar to the closest relative, Cocksfoot mottle virus, at the amino acid level.

Three contigs with a total of 3909 nt were recovered for the novel luteovirus. They contained a peptidase, RdRp, coat protein (Table 1) that were ~39%, 78 and 54% similar, respectively, to their closest relative at the amino acid level (Table 2).

## 4. Discussion

Metagenomic next-generation sequencing enables the unbiased discovery of viruses within samples that would otherwise be missed with conventional methods targeted to specific viruses. Using this approach, we report the discovery of a novel field vole paramyxovirus (UK*Ma* K4D) and a novel rabbit rotavirus A (L232). We also identifed a novel astrovirus, rotavirus A, in a field vole (UK*Ma*1) and a picorna-like virus in a rabbit (L232). While unbiased metagenomics is capable of identifying novel viral species, it is noticeable that only a small proportion of the total number of reads corresponded to viral reads, confirming previous observations [31]. An obvious limitation of metagenomics in detecting viral genomes is the requirement for a sufficient number of viral reads within a library to detect a virus and construct complete virus genomes, which is itself dependent on the RNA quality within a sample [32]. Comparison of virus abundance in each animal revealed an association between number of viral/host reads and sample quality, with rabbit L232 being the best quality sample. The difference between the quality of the rabbit and the field vole samples is likely due to the method of collection; the rabbit was euthanized and immediately processed [20], whereas the field voles were a result of routine pest control, and there was a delay between time of death and collection and sample preservation [11].

This study presents the first evidence of a narmovirus in European field voles. In every gene analyzed this virus clustered with narmoviruses that had been isolated from a range of rodents from diverse locations globally. Specifically, other viruses within this clade were identified in bank voles from Russia [33], wild rats in Australia [34] and common cane mice in Trinidad [35]. Such a phylogenetic pattern suggests that rodent narmoviruses are temporally and spatially widespread and arose from a single common ancestor, although this will need to be confirmed with a larger sample of viruses. We previously observed a similar pattern of shared ancestry for rodent alphacoronaviruses [25]. 

Phylogenetic analysis of the rabbit rotavirus A revealed both high sequence similarity (>96%) and clustering with rotavirus A sequences sampled from different mammalian species. However, the lack of a several reference sequences for all the segments analyzed, together with the ability of rotaviruses A to reassort [36], makes it difficult to draw any detailed conclusions about the origin of these viruses. In contrast, the rotavirus A identified in the UK field vole was considerably more divergent phylogenetically, falling in a more basal position, although with some similarity to another rodent rotavirus. In addition, the closest relatives only exhibited ~75% sequence similarity (Table 2), thus fulfilling the criteria for a novel rotavirus A genotype in each segment [37], and perhaps indicating the presence of another rodent-specific virus lineage. 

Although next-generation metagenomic sequencing has fundamentally changed virus discovery, many viruses still lack information about their host species [38] and the interpretation of results is greatly complicated by commonplace reagent contamination [39]. It is therefore important to confirm any metagenomic findings by PCR on the original samples. In addition, depending on the tissue, it may be difficult to determine the exact virus host (i.e., intestinal tissue sample) due to ‘contamination’ from the diet of the animal. Indeed, previous studies have shown the detection of viruses in gut and fecal samples of humans and animals that are likely derived from their diet [40,41,42]. Due to their detection in multiple animals we are confident that the field vole and the rabbit were the likely hosts for the paramyxovirus and the rotavirus A, respectively. However, it has not been determined whether the field vole is the true host of the rotavirus and astrovirus detected. The field vole-related astrovirus clustered with an astrovirus detected in fecal samples of European rollers (*Coracias garrulus*) in Hungary [43]. This bird species is found in a wide variety of habitats and is known to feed on insects and small vertebrates, including rodents [44]. However, as noted by the original authors, this virus shared a common ancestor with rodent-borne astroviruses so it is possible that it has a dietary origin since European rollers prey on small rodents [44]. Hence, the clade comprising UK*Ma*1 and Er/SZAL6/HUN/2011 astrovirus may in reality represent a rodent clade of mamastroviruses. This will require confirmation through the analysis of additional rodent astroviruses.

Another example of probable dietary contamination was the detection of the novel picorna-like virus and the plant viruses in the rabbit intestinal sample. According to our phylogenetic analysis, the host of the picorna-like virus was likely an invertebrate because it closely grouped with other invertebrate picorna-like viruses such as Beihai barnacle virus 4 (NC032446) and Beihai sesarmid crab virus 2 (NC032641). Dietary contamination was also the likely explanation for Blackbird arilivirus (NC040820) discovered in a metagenomic analysis of a mixed pool obtained from brain, lung and intestine tissues [45]. While the original authors suggested that the virus might have a plant origin, our phylogenetic indicates that both viruses in fact likely have an invertebrate host.

In sum, our results confirm that metagenomic next-generation sequencing is capable of discovering previously unknown and divergent RNA viruses. However, our data also highlight that definitive identification of the likely host species can be complicated because of dietary or environmental contamination. As a consequence, true assignment of host species requires wider prevalence studies together analysis of multiple tissue types.

## Figures and Tables

**Figure 1 viruses-12-00047-f001:**
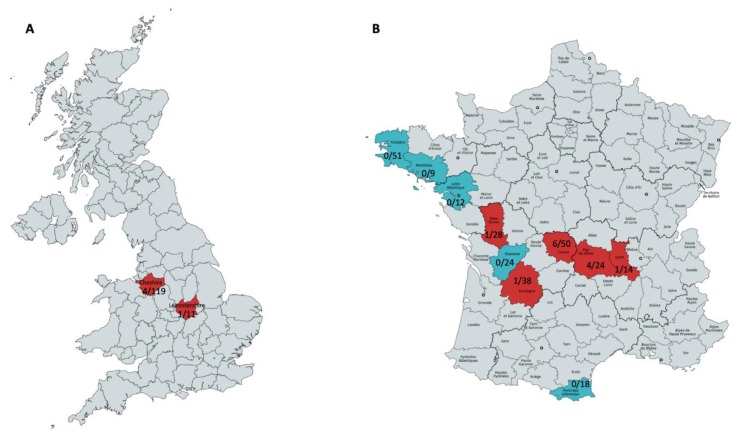
Maps of the UK (**A**) and France (**B**) illustrating the regions that the field vole paramyxovirus (**A**) and rabbit rotavirus A (**B**) were detected, along with their frequency. (**A**) Cheshire and Leicestershire (highlighted in red) had four and one positive field voles with paramyxovirus. (**B**) The departments Creuse, Puy-de-Dôme, Deux-Sèvres, Loire and Dordogne (highlighted in red) had six, four, one, one and one positives, respectively. The departments Charente, Finistère, Morbihan, Loire Atlantique and Pyrénées Orientales had no positives.

**Figure 2 viruses-12-00047-f002:**
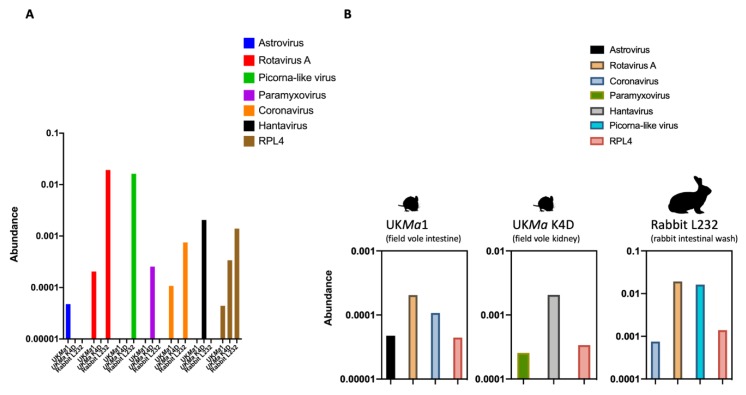
Overview of viral reads identified in this study. (**A**) Host associations (UK *Microtus agrestis* gut (UK*Ma*1), UK*Ma* K4D and rabbit L232) with the viral reads identified in this study: astrovirus, rotavirus A, picorna-like virus, paramyxovirus, coronavirus, hantavirus and host reference gene *ribosomal protein L4* (*RPL4*). (**B**) Individual representation of the abundance of viral reads in each animal sample, measured as a percentage of viral reads in log10 scale.

**Figure 3 viruses-12-00047-f003:**
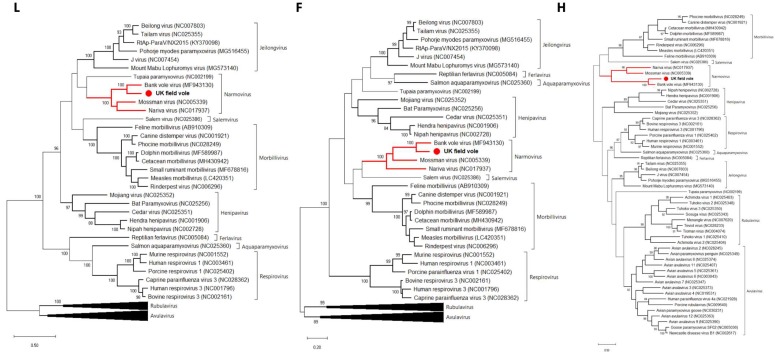
Phylogenetic relationships of the field vole paramyxovirus based on the *L*, *F* and *H* genes. The trees represent maximum likelihood phylogenetic analyses of paramyxovirus partial *L* (910 aa), *F* (242 aa) and *H* (205 aa) gene sequences, revealing that the rodent narmoviruses form a single clade within the *Paramyxoviridae*. The novel sequence obtained from the UK field vole UK*Ma* K4D was analyzed alongside reference sequences representing a diverse set of paramyxoviruses. Reference sequences are indicated by their GenBank accession numbers. Branch lengths are drawn to a scale of amino acid substitutions per site. Numbers above individual branches indicate bootstrap support; only values >80% are shown. Narmoviruses are highlighted in red and the novel UK field vole paramyxovirus is marked with a red dot. For clarity, some large viral clades have been collapsed.

**Figure 4 viruses-12-00047-f004:**
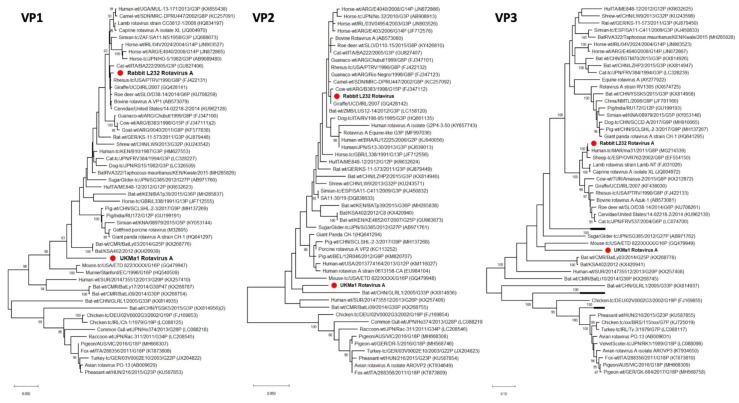
Phylogenetic relationships of the field vole and the rabbit rotavirus A based on the VP1, VP2 and VP3 segment. The trees represent maximum likelihood phylogenetic analyses of rotavirus A partial VP1 (648 aa), VP2 (329 aa) and VP3 (791 aa) segment sequences. The novel sequences obtained from the UK field vole UK*Ma*1 and the French rabbit L232 were analyzed alongside reference sequences representing rotaviruses A from different host species. Reference sequences are indicated by their GenBank accession numbers. Branch lengths are drawn to a scale of amino acid substitutions per site. Numbers above individual branches indicate bootstrap support; only values >80% are shown. The novel rotaviruses A are marked with a red dot. For clarity, some viral clades have been collapsed.

**Figure 5 viruses-12-00047-f005:**
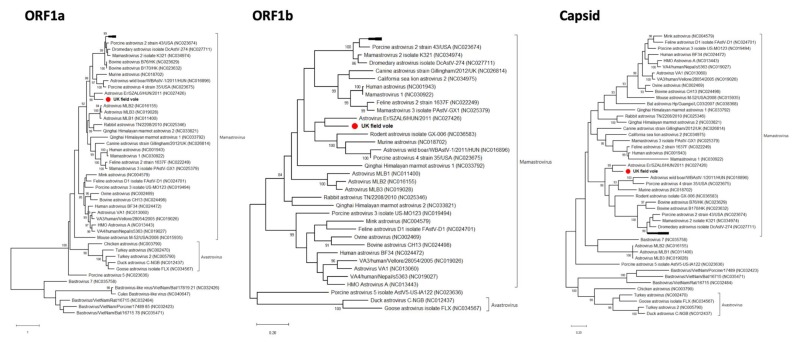
Phylogenetic relationships of the novel field vole astrovirus based on the *ORF1a*, *ORF1b* and *capsid* genes. The trees represent maximum likelihood phylogenetic analyses of astrovirus partial *ORF1a* (508 aa), *ORF1b* (298 aa) and capsid (350 aa) gene sequences. The novel sequence obtained from the UK field vole UKMa1 was analyzed alongside reference sequences representing all different genera of astroviruses. Reference sequences are indicated by their GenBank accession numbers. Branch lengths are drawn to a scale of amino acid substitutions per site. Numbers above individual branches indicate the bootstrap support; only values >80% are shown. The novel UK*Ma*1 astrovirus is marked with a red dot. For clarity, some viral clades have been collapsed.

**Figure 6 viruses-12-00047-f006:**
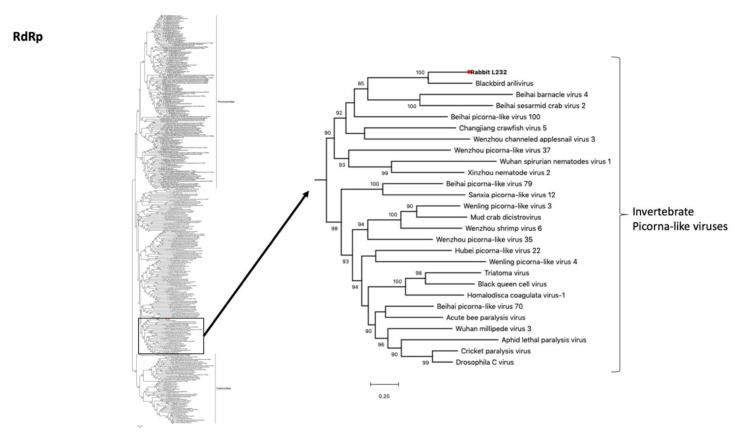
Phylogenetic relationships of the novel rabbit picorna-like virus based on analyses of the *RdRp* gene. Maximum likelihood phylogenetic analysis of picornavirus partial *RdRp* gene sequences, revealing that the rabbit picorna-like virus falls within a clade of invertebrate picorna-like viruses. The novel sequence obtained from the French rabbit L232 was analyzed alongside reference sequences representing different picorna-like viruses. Branch lengths are drawn to a scale of amino acid substitutions per site. Numbers above individual branches indicate the bootstrap support; only values >80% are shown. The novel rabbit L232 associated picorna-like virus is marked with a red dot.

**Table 1 viruses-12-00047-t001:** The samples and the corresponding viral hits including virus type, gene and size of the retrieved fragment utilized here.

Sample Name	Animal Species	Organ	Origin	Virus	Gene	Fragment Size (nt)	Accession Number
UK*Ma*1	*Microtus agrestis*	Intestine	UK	Rotavirus A	*VP1* (partial)	2145	MN626437
UK*Ma*1	*Microtus agrestis*	Intestine	UK	Rotavirus A	*VP2* (partial)	987	MN626438
UK*Ma*1	*Microtus agrestis*	Intestine	UK	Rotavirus A	*VP3* (partial)	2306	MN626439
UK*Ma*1	*Microtus agrestis*	Intestine	UK	Rotavirus A	*VP6* (partial)	878	MN626440
UK*Ma*1	*Microtus agrestis*	Intestine	UK	Rotavirus A	*NSP2* (partial)	900	MN626435
UK*Ma*1	*Microtus agrestis*	Intestine	UK	Rotavirus A	*NSP3* (partial)	349	MN626436
UK*Ma*1	*Microtus agrestis*	Intestine	UK	Astrovirus	*ORF1a* (partial)	1728	MN626433
UK*Ma*1	*Microtus agrestis*	Intestine	UK	Astrovirus	*ORF1b* (partial)	1554	MN626434
UK*Ma*1	*Microtus agrestis*	Intestine	UK	Astrovirus	Capsid (partial)	1052	MN626432
UK*Ma* K4D	*Microtus agrestis*	Kidney	UK	Paramyxovirus	*L* (partial)	2731	MN626428
UK*Ma* K4D	*Microtus agrestis*	Kidney	UK	Paramyxovirus	*H* (partial)	615	MN626427
UK*Ma* K4D	*Microtus agrestis*	Kidney	UK	Paramyxovirus	*F* (partial)	726	MN626426
UK*Ma* K4D	*Microtus agrestis*	Kidney	UK	Paramyxovirus	*M* (partial)	760	MN626429
UK*Ma* K4D	*Microtus agrestis*	Kidney	UK	Paramyxovirus	*P* (partial)	856	MN626431
UK*Ma* K4D	*Microtus agrestis*	Kidney	UK	Paramyxovirus	*N* (partial)	1046	MN626430
L232	*Oryctolagus cuniculus*	Intestinal wash	FR	Rotavirus A	*VP1* (partial)	3268	MN626420
L232	*Oryctolagus cuniculus*	Intestinal wash	FR	Rotavirus A	*VP2* (partial)	2645	MN626424
L232	*Oryctolagus cuniculus*	Intestinal wash	FR	Rotavirus A	*VP3* (partial)	2536	MN626421
L232	*Oryctolagus cuniculus*	Intestinal wash	FR	Rotavirus A	*VP6* (partial)	1336	MN626422
L232	*Oryctolagus cuniculus*	Intestinal wash	FR	Rotavirus A	*VP7* (partial)	978	MN626423
L232	*Oryctolagus cuniculus*	Intestinal wash	FR	Rotavirus A	*NSP2* (partial)	980	MN626417
L232	*Oryctolagus cuniculus*	Intestinal wash	FR	Rotavirus A	*NSP3* (partial)	951	MN626418
L232	*Oryctolagus cuniculus*	Intestinal wash	FR	Rotavirus A	*NSP4* (partial)	584	MN626419
L232	*Oryctolagus cuniculus*	Intestinal wash	FR	Picorna-like virus	Non-structural polyprotein (partial)	890	MN626416
L232	*Oryctolagus cuniculus*	Intestinal wash	FR	Picorna-like virus	Non-structural polyprotein contig 2 (partial)	2187	MN626416
L232	*Oryctolagus cuniculus*	Intestinal wash	FR	Picorna-like virus	Structural polyprotein (partial)	2349	MN626416
L232	*Oryctolagus cuniculus*	Intestinal wash	FR	Sobemovirus	Polyprotein P2ab (partial)	1653	MN626425
L232	*Oryctolagus cuniculus*	Intestinal wash	FR	Luteovirus	Peptidase (partial)	2067	MN626414
L232	*Oryctolagus cuniculus*	Intestinal wash	FR	Luteovirus	*RdRp* (partial)	1269	MN626413
L232	*Oryctolagus cuniculus*	Intestinal wash	FR	Luteovirus	Coat protein (partial)	573	MN626415

**Table 2 viruses-12-00047-t002:** Nucleotide and amino-acid identities of each gene/segment utilized in this study compared to the closest related viruses and the cut off for novel genotypes.

Virus	Gene/Segment	Animal	GenBank acc. No	Most Closely Related Virus	Nucleotide Sequence Identity (%)	Cut Off for Genotypes (%) *	Assigned Genotype
Rotavirus A	VP1	Rabbit	MN626420	Bovine rotavirus core protein	96.24	83	R2
		Field vole	MN626437	Human rotavirus A strain B10	76.96		R *
	VP2	Rabbit	MN626424	Rotavirus A giraffe/UCD/IRL/2007	98.38	84	C2
		Field vole	MN626438	Rotavirus A strain RVA/Cow-wt/ZAF/1605/2007/G6P [5]	78.21		C *
	VP3	Rabbit	MN626421	Rotavirus A isolate RVA/Human-tc/MAR/ma31/2011/G8P [14]	98.34	81	M2
		Field vole	MN626439	BatRVA/KEN/BATp39/Rousettus aegyptiacus/2015	69.74 (in aa)		M *
	VP6	Rabbit	MN626422	Rotavirus A strain RVA/roe deer-wt/SLO/D38-14/2014/G6P [15]	98.8	85	I2
		Field vole	MN626440	Human rotavirus A isolate Omsk07-79	78.46		I *
	VP7	Rabbit	MN626423	Rotavirus A strain RVA/Sheep-tc/ESP/OVR762/2002/G8P [14]	97.75	80	G8
	NSP2	Rabbit	MN626417	Rotavirus A strain RVA/Human-wt/ITA/111-05-27/2005/G6P [14]	96.99	85	N2
		Field vole	MN626435	Rotavirus A RVA/Dog-tc/JPN/RS15/1982/G3P [3]	76.01		N *
	NSP3	Rabbit	MN626418	Rotavirus A strain RVA/Human-wt/BEL/B10925/1997/G6P [14]	98.86	85	T6
		Field vole	MN626436	Rotavirus A RVA/Human-wt/JPN/12597/2014/G8P [14]	76.72 (in aa)		T *
	NSP4	Rabbit	MN626419	Rotavirus A strain dog-wt/GER/88977/2013/G8P1	98.43	85	E2
Astrovirus	*ORF1a*	Field vole	MN626433	Astrovirus Er/SZAL6/HUN/2011	46.3 (in aa)	-	-
	*ORF1b*		MN626434	Astrovirus Er/SZAL6/HUN/2011	70 (in aa)	-	-
	*Capsid*		MN626432	Astrovirus Er/SZAL6/HUN/2011	73.31 (in aa)	-	-
Picorna-like virus	*nonstructural polyprotein*	Rabbit	MN626416	Blackbird arilivirus	35.57 (in aa)	-	-
	*nonstructural polyprotein*		MN626416	Blackbird arilivirus	48.75 (in aa)	-	-
	*structural polyprotein*		MN626416	Blackbird arilivirus	34.24 (in aa)	-	-
Paramyxovirus	*L*	Field vole	MN626428	Bank vole virus	70.66 (in aa)	-	-
	*H*		MN626427	Bank vole virus	61.19 (in aa)	-	-
	*F*		MN626426	Bank vole virus	76.35 (in aa)	-	-
	*M*		MN626429	Bank vole virus	86.96 (in aa)	-	-
	*P*		MN626431	Bank vole virus	62.41 (in aa)	-	-
	*N*		MN626430	Bank vole virus	89.66 (in aa)	-	-
Luteovirus	*coat protein*	Rabbit	MN626415	Pea enation mosaic virus 1	53.90 (in aa)	-	-
	*Peptidase*		MN626414	Pea enation mosaic virus 1	38.88 (in aa)	-	-
	*RdRp*		MN626413	Alfalfa enamovirus 2	77.64 (in aa)	-	-

* Indicates novel unassigned genotype.

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
