# Peer review of "Discovery and Prevalence of Divergent RNA Viruses in European Field Voles and Rabbits"

_viruses, 2019, doi:10.3390/v12010047_

Round 1
Reviewer 1 Report
The authors report on the use of unbiased metagenomic virus discovery to identify novel RNA viruses in European field voles and rabbits.
The article is very well (extremely well) written; it stands out for clarity of presentation, logic, flow, and completeness throughout, from introduction to discussion.
Some minor questions for which responses from the authors are optional:
(1) Line 120 onwards: Contig confirmation PCR - How were the PCR amplicons sequenced?
(2) Figure 1: Identify RPL4 (write complete name)?
NOTE: Typographical error: Line 146: There are two periods at the end of the sentence.
Author Response
Very many thanks for handing our manuscript for Viruses so efficiently. We also thank the reviewers for their positive comments and suggestions.
For reviewer 1:
(1) Line 120 onwards: Contig confirmation PCR - How were the PCR amplicons sequenced?
We have added: “The PCR products were subjected to Sanger sequencing at Source Bioscience, Nottingham.”
(2) Figure 1: Identify RPL4 (write complete name)?
We have now provided the full name of the gene and the abbreviation in Italics in the legend to Figure 1.
NOTE: Typographical error: Line 146: There are two periods at the end of the sentence.
We have deleted the additional period.
Reviewer 2 Report
Manuscript No.: viruses-683329
Title: Discovery and prevalence of divergent RNA viruses in European field voles and rabbits
Authors: Tsoleridis et al
In this well-written study, the authors evaluated the virome of European field voles and rabbits, identifying a series of novel RNA viruses. In reality, the results are novel, interesting and well described so this reviewer has no major comments, criticisms.
Author Response
Very many thanks for handing our manuscript for Viruses so efficiently. We also thank the reviewers for their positive comments and suggestions.
Reviewer 2 did not request any changes in the manuscript.